# Development and characterization of rabbit monoclonal antibodies that recognize human spermine oxidase and application to immunohistochemistry of human cancer tissues

Armand W. J. W. Tepper[1], Gerald Chu[2], Vincent N. A. Klaren[1], Jay H. Kalin[2], Patricia Molina-Ortiz[3], Antonietta Impagliazzo[1]¤ *

1 Janssen Vaccine and Prevention, Leiden, The Netherlands, 2 Janssen Research & Development, Spring House, PA, United States of America, 3 Precision for Medicine, (UK labs), Royston, United Kingdom

¤ Current address: Genmab B.V. Uppsalalaan, Utrecht, The Netherlands
* anim@genmab.com

## Abstract

The enzyme spermine oxidase (SMOX) is involved in polyamine catabolism and converts spermine to spermidine. The enzymatic reaction generates reactive hydrogen peroxide and aldehydes as by-products that can damage DNA and other biomolecules. Increased expression of SMOX is frequently found in lung, prostate, colon, stomach and liver cancer models, and the enzyme also appears to play a role in neuronal dysfunction and vascular retinopathy. Because of growing evidence that links SMOX activity with DNA damage, inflammation, and carcinogenesis, the enzyme has come into view as a potential drug target. A major challenge in cancer research is the lack of characterization of antibodies used for identification of target proteins. To overcome this limitation, we generated a panel of high-affinity rabbit monoclonal antibodies against various SMOX epitopes and selected antibodies for use in immunoblotting, SMOX quantification assays, immunofluorescence microscopy and immunohistochemistry. Immunohistochemistry analysis with the antibody SMAB10 in normal and transformed tissues confirms that SMOX is upregulated in several different cancers. Together, the panel of antibodies generated herein adds to the toolbox of high-quality reagents to study SMOX biology and to facilitate SMOX drug development.

## Introduction

The polyamine metabolic pathway is finely tuned by the combined action of multiple enzymes which determine polyamine (PA) levels. PAs are involved in many cellular functions such as cell growth, proliferation, differentiation, migration, gene regulation and in the maintenance of cellular oxidative homeostasis [1]. Deregulation of polyamine metabolism is known to be associated with several epithelial cancers [2,3] as well as with various neurodegenerative

**Data Availability Statement:** All relevant data are within the paper and its Supporting Information files.

**Funding:** The author(s) received no specific funding for this work.

**Competing interests:** The authors have declared that no competing interests exist.

diseases such as Alzheimer's disease [4], Parkinson's disease [5], diabetic retinopathy [6–8], traumatic brain injury [7,9], and ischemic brain damage [10].

Alterations in polyamine metabolism may cause cellular damage and cell death through the generation of oxidative byproducts derived from polyamine catabolism. Two pathways are primarily responsible, the first involves a two-step process where both spermidine (Spd) and spermine (Spm) are first acetylated in their *N*1 positions by SSAT [11] to form *N*1-acetylated PAs which are either excreted from the cell through their specific transport system, or oxidized by human *N*1-acetylpolyamine oxidase (hPAOX) in the peroxisome, resulting in the formation of $H_2O_2$, 3-acetoamidopropanal, and putrescine (Put) or Spd. The second pathway is a one-step reaction where Spm is directly converted to Spd via human Spermine oxidase (hSMOX; previously PAOh1) [12]. hSMOX has been observed in the cytoplasm and nucleus [13] where it can produce Spd, $H_2O_2$, and 3-aminopropanal (3-AP) [14]. Both hSMOX and hPAOX are flavin (FAD) containing enzymes with 40% sequence identity.

In contrast to hPAOX, hSMOX is highly induced by inflammatory cytokines [15], and several reports suggest that hSMOX, but not hPAOX, is critically involved in chronic inflammation [15,16], cancer [16–18], neuronal dysfunction, and vascular defects in the retina [6,8]. Dysregulation of hSMOX not only causes changes in cellular polyamine levels but also in the levels of its byproducts—$H_2O_2$ and acrolein—that are both reactive and may reach critical concentrations that result in cellular damage and cancer initiation [19]. Treatment with MDL 72527, a polyamine oxidase inhibitor, results in reduced spermidine levels, decreased inflammation and DNA damage, decreased proliferation, and reduced tumor numbers in several *in vivo* models [3]. Treatment with MDL 72527 significantly improved ERG responses in diabetic retinas, and inhibited diabetes-induced retinal ganglion cell damage and neurodegeneration [20]. Consequently, hSMOX has been suggested as an attractive therapeutic target in cancer interception, in diabetes-induced retinal neurodegeneration and in visual dysfunction [6,21].

At present, commercially available antibodies against hSMOX are polyclonals with limited validation. The lack of specific monoclonals against hSMOX for different applications hampers hSMOX research and efforts to validate hSMOX as a viable and tractable therapeutic target. Thus, we have generated and characterized a series of specific and selective rabbit monoclonals against human and murine SMOX for different applications, including immunohistochemistry of human tissues.

## Materials and methods

### Peptides and proteins

Immunization peptides were synthesized by InnoPep (San Diego, USA) under the supervision of ExonBio (San Diego, USA). A N- or C-terminal cysteine was introduced to all hSMOX peptide sequences to allow conjugation to keyhole limpet haemocyanin (KLH) employing sulfo-succinimidyl 4-[*N*-maleimidomethyl]cyclohexane-1-carboxylate (Sulfo-SMCC) chemistry. All peptides had the expected size and were >95% pure as checked by LC/MS. A detailed protocol describing rhSMOX production will be published elsewhere (manuscript submitted). Briefly, hSMOX constructs were produced in E.coli BL21(D3) transformed with the plasmid PET28b (+) containing the hSMOX isoform 1 gene (UniProtKB—Q9NWM0) or variants thereof. All rhSMOX variants contained an N-terminal $His_6$-tag to facilitate purification. The proteins were purified from cell lysates by sequential chromatography using Ni-sepharose Excel (GE cat# 17371201) using a gradient from 5 mM to 0.5 M imidazole in 10 mM HEPES, pH7.4, and CaptoQ Impres (GE cat# 17-5470-55) using a gradient from 0 to 0.5 M NaCl in 10 mM HEPES, pH 7.1. All employed proteins were >95% pure and had the expected size as judged by chip-based electrophoresis (Bioanalyzer 2100, Agilent). The rhSMOX variants contained

FAD as judged by the 450 nm absorption and were catalytically active as judged by the production of $H_2O_2$ after incubation of 20nM rhSMOX with 100uM spermine substrate for 30 min (10mM HEPES, 150mM NaCl, 0.1% BSA, pH 7.4) as measured using the HyperBlu reagent (Lumigen cat# HPB-00005) that provides a luminescent signal after reaction with $H_2O_2$.

## Monoclonal production

Immunizations were performed by Pacific Immunology (Ramona, USA) in accordance with the Animal Welfare Act under the supervision of ExonBio (San Diego, USA). Rabbit immunization was carried out in strict accordance with the recommendations in the Guide for the Care and Use of Laboratory Animals of the National Institutes of Health. The project was approved by designated member review and follows rabbit antibody production protocol (Protocol 1). This protocol is approved by the Institutional Animal Care and Use Committee. Immunizations and bleeds are considered to cause minimal pain and suffering, and no anesthesia or analgesia is required. At the completion of the study, euthanasia of the rabbits is performed with intravenous injection of Euthasol euthanasia solution, following the manufacturer's and veterinary recommendations of a dosage of 1 ml for each 10 lbs. (4.5 kg) of body weight. Euthanasia by intravenous injection of an overdose of pentobarbital sodium in combination with phenytoin sodium is consistent with the recommendations of the Panel on Euthanasia of the American Veterinary Medical Association.

Minimally, 8 week old female New Zealand White Rabbits were immunized with 0.2 mg antigen in Complete Freund's Adjuvant (CFA) by subcutaneous injection in the loose skin behind the neck. The immunization was repeated three times three weeks apart with the appropriate immunogen in Incomplete Freund's Adjuvant (IFA). After 11 weeks, target specific Ab titers were determined by serial dilution ELISA using rhSMOX coated plates, after which the animals were euthanatized, and the spleen was harvested. Splenocytes were prepared by Ficoll-Paque Plus gradient centrifugation. Antigen-specific single plasma cells were isolated by FACS sorting with antigen specific fluorescence probes and dispensed to 96-well PCR plates at one cell per well. Antibody V-regions of heavy and light chains were reverse-transcribed and PCR amplified. Amplified V-regions were further assembled to an expression unit with constant regions of the heavy and light chains, as well as a CMV promoter and a bGH terminator. The expression unit was transfected into HEK293F cells for transient expression of the full-length rabbit antibody. Expressed supernatants were further screened with ELISA. ELISA positive samples were further cloned and antibodies were produced.

## SMAB expression and purification

SMAB cloning and expression was performed by ExonBio (San Diego, USA). The heavy and light-chain CDR coding regions of SMABs were respectively cloned into pRab293H2 and pRab293L3 vectors and transfected to Expi293F cells (A14527, ThermoFisher) using an Expi-Fectamine293 transfection kit (A14524, Thermofisher). Transfected cells were grown in Expi293 medium (A1435101, ThermoFisher). Supernatants were harvested when the viability was around 50% (day 5–7 post-transfection). For SMAB purification, 30 ml cell cultures were first clarified by centrifugation (15,000rpm, 20min) after which 1 ml equilibrated protein A resin (cat# 17513801, Cytiva) was added. The mixture was equilibrated for at least 2 hours at 4˚C with rolling, after which SMAB-bound resin was washed with PBS by centrifugation (1500rpm, 10min). The resin was transferred to a 5 ml column, washed with 60 ml PBS, and eluted with 0.1 M citrate buffer, pH 3.0, by gravity. The fractions were collected at 0.5 ml per vial and neutralized with 0.05 ml 1 M Tris-HCl, pH 8.0. Protein content was measured by OD280 and peak fractions were pooled and dialyzed against PBS at 4˚C using a membrane

with 30 kDa cut-off. Antibody concentrations were measured by OD280 on spectrophotometer with coefficient adjustment and purity was checked by SDS-PAGE. Proteins were stored at −80˚C.

## Epitope binning

Epitope binning was performed on an Octet HTX machine (ForteBio) employing black 96-well plates (3694, Costar) using an in-tandem assay format. To allow immobilization of rhSMOX, the protein was first biotinylated using the EZ-Link NHS-PEG4 biotinylation kit (21455, ThermoFisher) according to the manufacturer's instructions. All Octet solutions were made up in ForteBio kinetic buffer (PN 18–1105, ForteBio). Biotinylated rhSMOX (50nM) was immobilized on Octet streptavidin (SA) sensors (18–5019, ForteBio) for 600s followed by incubation with SMAB solution 1 (50nM) for 600s to reach saturation, and then the capacity of a different SMAB to bind to the rhSMOX/SMAB complex was measured by incubation of SMAB solution 2 (50nM) for 300s. Data were analyzed with the Octet software.

## Activity assay

The capacity of the SMABs to inhibit rhSMOX activity was measured by first incubating 4 ul 6nM rhSMOX with 4 ul 300 nM SMAB in a white 384-well assay plate (6057480, PerkinElmer). After 30 min preincubation, 4 ul 4mM rhSMOX spermine substrate was added and incubated for another 30 min. The $H_2O_2$ produced was measured by the addition of 6 ul HyperBlu reagent (HPB-0005, Lumigen), incubating for 30 min, and measuring luminescence on a Synergy Neo plate reader. All solutions were made up in 10 mM HEPES, pH 7.4, containing 0.01% BSA, 0.05% Tween-20 and 150 mM NaCl.

## Cell culture and lysis

A549 cells were cultured using DMEM (Gibco) supplemented with 10% Fetal Calf Serum at 37˚C and 5% $CO_2$. To stimulate hSMOX expression, the medium was supplemented with 10 µM BENSpm (Tocris, 0468) for 24 h. For the preparation of cell lysates, cells were washed twice with cold PBS (Gibco), harvested either with a cell lifter (Millipore, CLS3008) or by the addition of 2 mL TrypLE Select (Gibco, 12563011) per flask and incubation for 5 min at 37˚C. Cells were pelleted at 300 x $g$ for 5 min, washed, flash frozen and stored at −80˚C. Lysates were prepared by resuspending pellets (~5M cells) in 200 µL cold RIPA buffer (Thermo, 89900) supplemented with 5X Protease Inhibitor Cocktail + 1X EDTA (Thermo, 87786) and incubated on ice for 15 min. Solutions were clarified by centrifugation at 16,000 x $g$ at 4˚C for 10 min and protein concentration was determined using a BCA Protein Assay Kit (Thermo, 23227).

## AlphaLISA

The AlphaLISA assay was setup by identifying suitable antibodies and optimizing antibody concentrations according to the manufacturer's instructions (see results for details). For the measurement of [SMOX] in A549 cells, cell pellets containing 6 million cells as determined from automated trypan blue cell counting using a Luna (Logos Biosystems) cell counter were resuspended and lysed in 500 µl AlphaLISA buffer followed by centrifugation at 14,000 g. The supernatant was aliquoted and stored at −80 ˚C until use. In the final assay, a SMAB2/ His6-tagged Fab33 mixture was prepared by diluting SMAB2 to 0.5 nM and and His6-Fab33 to 5 nM in 1x AlphaLISA buffer. A volume of 5 µL of this mixture was added to 10 µL of 10x diluted lysate sample (for determinations of SMOX concentration) or 10 µl AlphaLISA buffer containing a known amount of SMOX (for calibration) in a 384 well white assay plate

(OptiPlate-384, Perkin Elmer) and incubated for one hour at RT. Then 5 μL 50 μg/ml acceptor beads (AL178C, Perkin Elmer) was added followed by incubation for one hour at RT. For the last step of the assay, 5 μL donor beads (200 μg/mL) (AS105D, Perkin Elmer) was added per well. After one hour incubation at RT in the dark, the Alpha signal was read using an Alpha compatible reader (Synergy NEO, BioTek). Lysate and calibration samples were measured in duplicate.

## Western blot

Proteins and cell lysates were denatured by heating to 95°C for 5 min in LDS Sample Buffer (Thermo, NP0008) supplemented with 1X reducing agent (Thermo, NP0009), resolved using 4–12% Bis-Tris Protein Gels (Thermo, NP0329BOX) run in MES SDS Running Buffer (Thermo, NP0002), and transferred to nitrocellulose membranes (Thermo, IB23002) using the iBlot 2 (Thermo). Membranes were blocked with Odyssey Blocking Buffer in TBS (LI-COR, 927–50000) for 2 h at RT and incubated with primary antibodies in Odyssey Blocking Buffer + 0.2% Tween-20 (Sigma, P9416) overnight at 4°C. Membranes were washed three times with 1X TBST (G-Biosciences, R042) and incubated with secondary antibodies for 2 h at RT. After washing three times with 1X TBST, blots were imaged with the LI-COR Odyssey (LI-COR Biosciences, Lincoln, NE). Antibodies and reagents: GAPDH (Thermo, AM4300, 1:5000, 0.25 μg/mL), goat anti-rabbit (LI-COR, 926–32211, 1:15000), goat anti-mouse (LI-COR, 926–68020, 1:15000), Chameleon Duo Pre-Stained Protein Ladder (LI-COR, 928–6000).

## Immunofluorescence

Cells were seeded at 10,000 and 15,000 cells per well in 96 well clear bottom imaging plates (353219, Falcon). After 24h, the medium was refreshed with plain medium or medium containing 10 μM BENSpm. After another 24h, cells were washed with PBS and fixed for 5 minutes using 10% neutral buffered formalin (HT5011, Sigma). After removing the formalin cells were washed with PBS followed by a blocking/permeabilization step using PBS 1% Bovine Serum Albumin, 0.1% Triton X-100 (staining buffer). Cells were stained by incubation with various anti-SMOX antibodies at 0.1–3 ug/mL for one hour. Primary antibodies were removed by washing the wells three times with 150 μL PBS, 0.1% Triton X-100 (wash buffer). Secondary staining was achieved by incubating the cells with 1 ug/mL AlexaFluor 488 labeled anti-rabbit IgG (A11034, Invitrogen) and 10 ng/mL DAPI in staining buffer. After one hour the wells were washed three times with 150 μL wash buffer followed by the addition of 100 μL PBS per well. The wells were imaged using a High Content imager (Pathway 855, Becton Dickinson) with a 20x objective taking 4 images per well per channel. Alexa Fluor 488 and DAPI images were stitched and overlaid using ImageJ software (version 1.53g10).

## Immunohistochemistry

All IHC experiments and tissue scorings were performed by Precision Medicine (Royston, UK). For the optimization of the IHC conditions to detect hSMOX using SMAB antibodies, positive and negative cell line controls were chosen based on the literature, the western blot and immunofluorescence results. Raji with a low basal hSMOX expression, A549 and BENSpm stimulated A549 cell lines were used as negative, low- and high-expressing controls, respectively. Cell pellets of the indicated cells were formalin-fixed, paraffin-embedded (FFPE) to create a 1.5 mm core tissue microarray (SMOX-TMA). Optimized IHC conditions for the SMAB10 antibody were further validated on full-face sections of xenograft samples (two Raji and two A549 xenografts). For the validation of SMAB10 IHC conditions on human samples, nine full-face tumour and two multitumour TMAs were purchased from commercial

biobanking services (BioIVT and US Biomax). Full-face sections included carcinomas of different origins including colon, stomach, lung, liver, pancreas and prostate. TMAs covered a core of tumors and normal tissues of multiple origins including breast (normal 6; ID carcinoma 48), colon (normal 11; adeno 47), lung (normal 9; squamous 30; adeno 19), and prostate (normal 10; adeno 47). Tissue microarray cores were excluded from the analysis if they were washed off, did not contain tumor, or the sample was not evaluable for other reasons. For each IHC assay, colon FFPE sections were prepared as indicated below, and stained as assay control sample using anti-vWF antibody, validating the detection reagents. Additionally, SMOX-TMA sections were used as SMAB antibody control of the immunoreactivity consistency and to facilitate the comparison between assays. From each FFPE sample, serial sections of 4 μm were cut, mounted onto Fisherbrand Superfrost Plus charged slides, and placed in an oven at 37˚C (± 2˚C) for 16 h. Sections were treated with Envision FLEX 3-in-1, High pH heat-mediated antigen retrieval solution (Dako, Glostrup, Denmark) for 20 min at 98˚C min, for deparaffination and antigen retrieval, using a PT Link (Agilent Technologies, Santa Clara, CA). Immunohistochemistry was performed using a Dako PLUS autostainer (Agilent Technologies, Santa Clara, CA) and EnVision FLEX Kit, High pH (Link) reagents (K802321-2, Agilent). Briefly, sections were incubated with peroxidase blocking (Kit) for 10 minutes, follows by serum-free protein blocking (Cat.X0909, Agilent) for 10 minutes. Anti-SMOX primary antibodies were applied for 60 minutes at ambient temperature, with antibody concentrations ranging from 10 to 2.5 μg/mL, after which sections were rinsed with wash buffer (kit). Envision FLEX HRP-Polymer (kit) was subsequently applied for 30 minutes. After washing with wash buffer, sections were incubated with DAB HRP-substrate (kit) for 10 min. The reaction was stopped with distillate water and immunostained slides were counterstained with haematoxylin, dehydrated in ascending alcohol, ending in xylene, and coverslipped using a non-aqueous mounting medium and standard glass coverslips.

Immunostained slides were scanned using Aperio AT turbo scanner (Leica Biosystems, Nussloch, Germany). Individual patient informed consent had been obtained for all human samples.

## Statistical analysis

All analysis were performed using Graphpad Prism version 8. Significance levels of SMAB inhibitory data (Fig 2B) were determined by ordinary one-way ANOVA without matching or pairing using Dunnet statistical hypothesis testing, a 95% confidence interval, and using the activity data of the control (no SMAB) as the reference in multiple comparisons. Curve analysis for the AlphaLISA rhSMOX assay calibration line was performed by fitting unweighted data to a 3-parameter logistic curve (3-PL) $Y = B+X^*(T-B)/(EC50+X)$ where Y is the experimental signal, X is the rhSMOX concentration in ng/ml, B is the signal obtained in absence of rhSMOX, T is the signal at full saturation with rhSMOX, and EC50 is the half maximal effective concentration corresponding to the curve inflection point (S1 Table). Confidence intervals of EC50 values (alpha = 0.05) were estimated using profile likelihood. All numerical operations were performed in Microsoft Excel for Windows 10.

## Results

### Commercial antibody evaluation

Before engaging in the development of hSMOX specific antibodies, we evaluated three commercially available affinity-purified polyclonal antibody preparations obtained from Sigma-Aldrich (HPA047117, HPA060198, SAB1101510) in Western Blot (WB) and Immunofluorescence (IF) experiments. For WB, we utilized catalytically active full length recombinantly

expressed hSMOX (rhSMOX), as well as lysates of human lung carcinoma cells (A549). A549 cells endogenously produce relatively high basal levels of hSMOX [22], which can be further amplified to reach >5x the baseline expression by treatment with BENSpm [12,23]. Raji, U937, T47D and HepG2 cells with a low basal hSMOX expression [22] were included as negative controls. Only HPA060198 intensely stained rSMOX (S1 Fig). This antibody also stained non-stimulated hSMOX in A549 extracts (65 kDa), with approximately 4-fold stronger staining for BENSpm induced A549 cells. Very faint hSMOX bands were observed for SAB1101510. Curiously, HPA060198 and SAB1101510 stained a protein running immediately below hSMOX (~62 kDa) in all cell types used. The intensity of this band is not dependent on BENSpm stimulation and is also observed in hSMOX negative cells. The origin of this band has not been investigated further but may correspond to a different protein as it is not recognized by any of the mAbs we generated (see below). All three antibodies also provided several bands at lower MW, as might be expected considering the polyclonal nature and the relatively long peptide sequences used for immunization. In immunofluorescence experiments performed using native and BENSpm treated A549 cells (see also below) with different dilutions of primary antibody, only SAB1101510 produced a staining signal, however it did not increase with BENSpm stimulation (S2 Fig) making it likely that aspecific binding is responsible for most of the signal intensity. Thus, we surmised that the commercial polyclonal antibodies lacked the specificity and sensitivity required for our purposes.

## Antigen design

To guide the selection of immunization peptides, we used a publicly available hSMOX homology model [24] generated from the published structure of murine polyamine oxidase (mPAOX; PDB ID: 5MBX) and the hSMOX isoform 1 sequence (Swissprot ID Q9NWM0). The N-terminal sequence is missing from the hSMOX model as it was not resolved in the mPAOX template structure. Five different peptide sequences were selected based on their predicted solvent exposure and similarity with hPAOX (Figs 1 and S3). The C-terminal (CT) and N-terminal (NT) peptide sequences are highly conserved in SMOX from different organisms. The LL1 and LL2 peptide sequences are part of the long loop (LL) occurring in hSMOX but not in hPAOX, and the LL2 sequence was previously used to generate a rabbit polyclonal [25] that has been used in multiple works (e.g. [26]). None of the peptides have notable homology with hPAOX (S3 Fig). To generate conformational mAbs we also immunized with full length recombinantly expressed rhSMOX.

## Antibody production and screening

Two rabbits were immunized with each of the five KLH-conjugated peptides, as well as with full-length recombinant hSMOX (rhSMOX). All rabbit sera produced a strong immunogenic response as detected using antigen-coated ELISA plates (not shown). Rabbits immunized with peptide LL1 and full length rhSMOX were selected for further mAb development. For each selected rabbit, 192 hybridoma supernatants were screened by ELISA for reactivity against LL1 and rhSMOX. Respectively, for rabbits immunized with LL1 and rhSMOX, 11 and 22 clone supernatants that demonstrated potent binding were selected for follow-up. The mAb genes were cloned, expressed, and purified. In ELISA, all but one (#117) clone bound to rhSMOX with double-digit pM EC50 values (Table 1), in line with the high affinities reported for other rabbit antibodies [27]. We found that 21/22 clones were reactive towards rhSMOX but not towards LL1, while one clone recognized both antigens. Next, to be able to distinguish long-loop binders from mAbs binding to other parts of rhSMOX, we generated a folded and catalytically active recombinant rhSMOXΔLL mutant that does not contain the LL1 and LL2

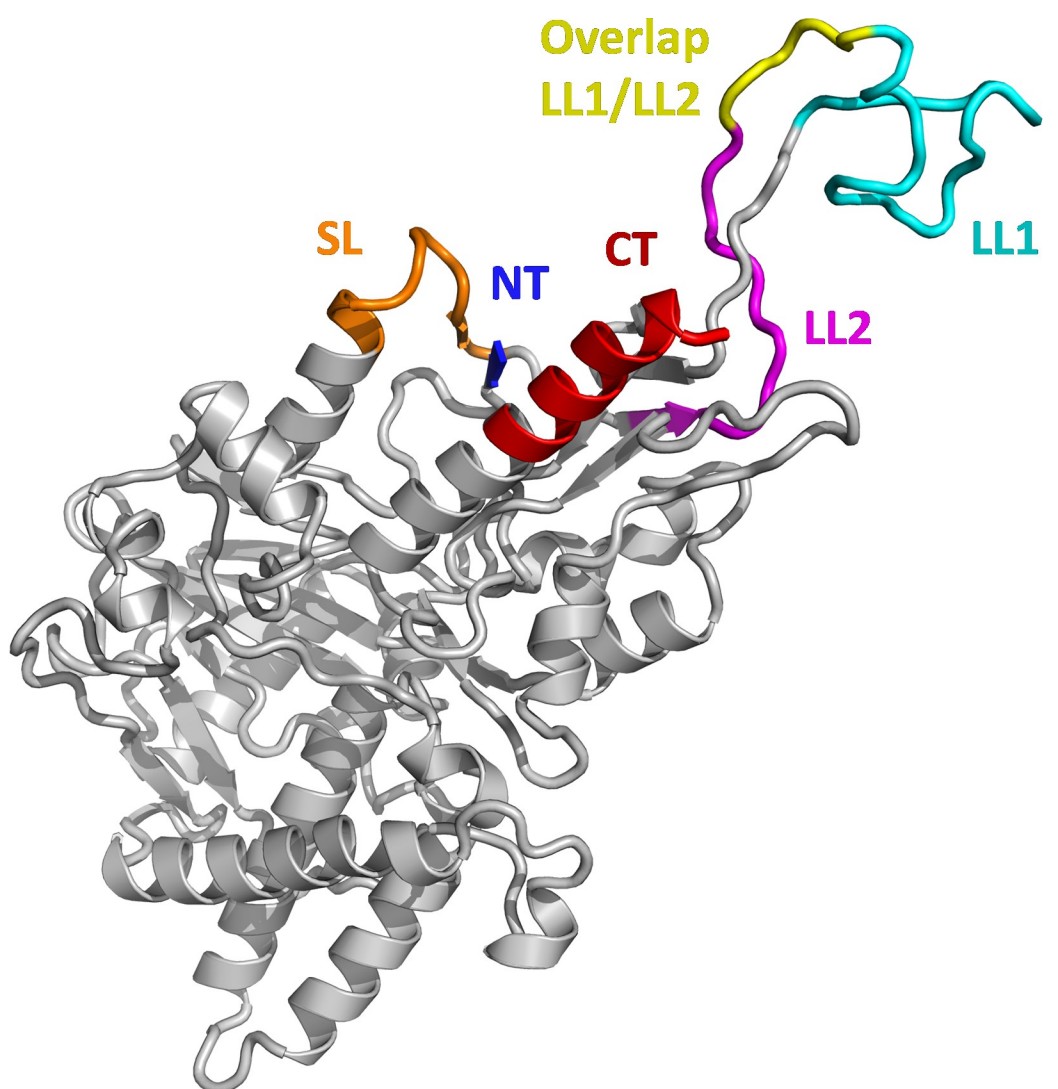

| Peptide | Sequence |
|---------|----------|
| NT | ESSGDSADDPLSRGC |
| SL | AEGIPAHVIQLGC |
| LL2 | CIHWDQASARPRGPEIEPR |
| LL1 | PEIEPRGEGDHNHDTGEGGQGGEEPRGGRWC |
| CT | CARLIEMYRDLFQQGT |

**Fig 1. Immunization peptides.** Structural model of human SMOX and immunization peptides.

sequences (Δ286–317). All selected clones from the LL1 immunization were positive for rhSMOX but not for rhSMOXΔLL, while all clones from the rhSMOX immunization recognized both rhSMOX and rhSMOXΔLL. Because peptides LL1 and LL2 have 6 amino acid

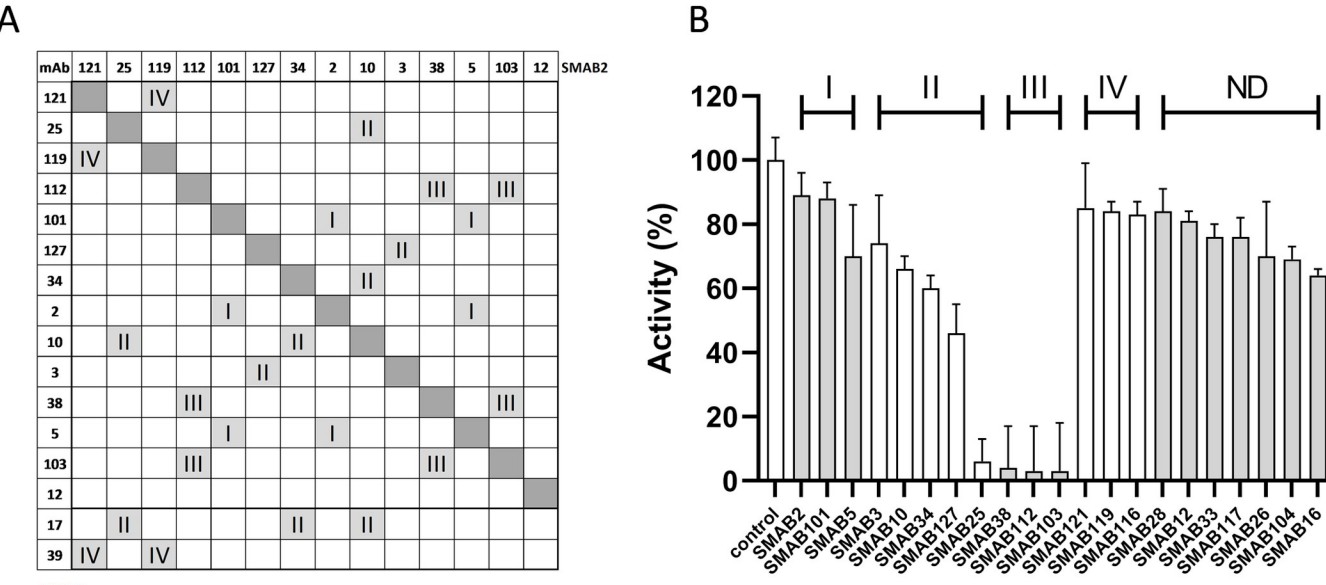

**Fig 2. Epitope binning and enzymatic activity inhibition.** (A) Epitope binning. The SMAB families identified by epitope binning are indicated with roman numerals. SMAB17 and SMAB39 were tested in one direction only. (B) SMAB inhibition of rhSMOX activity. Activity data were normalized against the control. Data bars represent the average ± std of quadruplicate measurements. With the exception of SMAB 2 and 101, differences for all SMABs were significant with $p < 0.05$ with respect to the non-inhibited control.

residues (PEIEPR) in common, the 11 selected clones from the LL1 immunization were also tested for reactivity against the LL2 peptide. We found that only one clone was selective for LL1 and that 10/11 clones bound to both LL1 and LL2 with similar affinities, indicating that these mAbs bind to epitopes within the sequence PEIEPR. Similarly, clones from the rhSMOX immunization were also tested for reactivity towards immunization peptides LL1, LL2, SL, NT and CT, allowing us to establish that 5/22 clones bound to epitopes within the C-terminal peptide CT, 1/22 bound to both LL1 and LL2, while none bound to SL or the N-terminal peptide NT. Eventually, 4 clones from the LL1 immunization and 20 clones from the rhSMOX immunization were selected for larger scale mAb expression and purification. These mAbs will be referred to as SMABs in the remainder of the text.

## SMABs epitope characterization

To establish which SMABs bind to epitopes in close vicinity, epitope binning experiments were performed with a selection of 16 purified SMABs using the Bioforte Octet platform. In this experiment, one SMAB (SMAB1) is first allowed to bind to immobilized rhSMOX, and then the capacity of another SMAB (SMAB2) to bind to the rhSMOX/SMAB1 complex is measured, where the absence of binding signals suggests similar or adjacent epitopes on rhSMOX. All tested SMABs bound to immobilized rhSMOX. We were able to identify four families I-IV of SMABs in which strong (<20% binding amplitude vs control) mutual binding competition is observed (Fig 2). No competition was observed for SMAB12 indicating that it binds to an epitope unique among all SMABs tested. SMAB3 binds to the CT peptide and was grouped into family II, thus SMAB28 that also binds to the CT peptide and was not tested in epitope binning can also be assigned to this family. Similarly, it is likely that SMABs in family II that did not bind the linear CT peptide (SMAB# 10, 17, 25, 34) bind close to the CT domain of the enzyme. To establish which SMABs were inhibiting the enzymatic activity of rhSMOX, we performed activity assays measuring production of the $H_2O_2$ reaction co-product formed during

Development and characterization of rabbit monoclonal antibodies that recognize human spermine oxidase

**Table 1. Overview of SMAB properties.**

| mAb clone | Immunogen | ELISA EC50 rSMOX (pM) | ELISA reactivity peptides LL1 | LL2 | SL | NT | CT | rSMOX rhMOX | rhSMOXALL | SMOX activity inhibition | Epitope binning family | WB rhSMOX | rmSMOX | rhPAO | A549 native SMOX | A549 stimulated SMOX | A549 background | RAJI SMOX | Fluor. A549 native intensity | A549 stimulated intensity | Intensity ratio stim/native | IHC A549 native intensity | A549 stimulated intensity | RAJI native intensity | Binding epitope |
|---|---|---|---|---|---|---|---|---|---|---|---|---|---|---|---|---|---|---|---|---|---|---|---|---|---|
| SMAB2 | rSMOX | 27 | - | - | - | - | - | + | + | - | I | - | - | - | - | - | - | - | + | ++ | ++ | ND | ND | ND | |
| SMAB3 | rSMOX | 14 | - | - | - | - | + | + | ND | + | II | ++ | + | - | + | ++ | +/- | - | + | +++ | ++ | ++ | ++ | ++ | C-terminal peptide |
| SMAB5 | rSMOX | 21 | - | - | - | - | - | + | ND | + | I | - | ND | - | - | - | - | - | + | +++ | +++ | ND | ND | ND | |
| SMAB10 | rSMOX | 16 | - | - | - | - | - | + | ND | + | II | + | + | - | +/- | + | + | - | + | +++ | ++ | - | - | - | C-terminal region |
| SMAB12 | rSMOX | 20 | - | - | - | - | - | + | + | - | ND | - | ND | - | - | - | + | - | + | +++ | ++ | ND | ND | ND | |
| SMAB16 | rSMOX | 25 | - | - | - | - | - | + | ND | + | (II) | - | ND | - | - | - | + | - | - | + | + | ND | ND | ND | (C-terminal region) |
| SMAB17 | rSMOX | 23 | - | - | - | - | - | + | ND | ND | II | ++ | + | - | - | - | - | - | ++ | +++ | ++ | - | + | - | C-terminal region |
| SMAB25 | rSMOX | 22 | - | - | - | - | - | + | ND | ++ | II | - | ND | - | - | - | + | - | - | ++ | ++ | ND | ND | ND | C-terminal region |
| SMAB26 | rSMOX | 23 | - | - | - | - | - | + | + | + | (II) | + | + | - | - | - | + | - | + | ++ | ++ | ND | ND | ND | (C-terminal region) |
| SMAB28 | rSMOX | 25 | - | - | - | - | + | + | + | - | (II) | + | + | - | - | + | ++ | - | + | +++ | + | ND | ND | ND | C-terminal peptide |
| SMAB33 | rSMOX | 30 | - | - | - | - | - | + | + | + | ND | - | - | - | - | - | + | - | + | ++ | + | ND | ND | ND | C-terminal region |
| SMAB34 | rSMOX | 44 | - | - | - | - | - | + | ND | + | II | + | + | - | - | - | + | - | + | +++ | +++ | - | -/+ | - | C-terminal region |
| SMAB38 | rSMOX | 19 | - | - | - | - | - | + | ND | ++ | III | - | ND | - | - | - | + | - | - | ++ | ++ | ND | ND | ND | (active site) |
| SMAB101 | rSMOX | 45 | - | - | - | - | - | + | ND | - | I | - | ND | - | - | - | - | - | + | ++ | +++ | ND | ND | ND | (active site) |
| SMAB103 | rSMOX | 35 | - | - | - | - | - | + | ND | ++ | III | - | ND | - | - | - | + | - | - | ++ | +++ | ND | ND | ND | (active site) |
| SMAB104 | rSMOX | 28 | - | - | - | - | - | + | ND | + | ND | - | ND | - | - | - | + | - | - | - | - | ND | ND | ND | |
| SMAB112 | rSMOX | 28 | - | - | - | - | - | + | + | ++ | III | - | ND | - | - | - | - | - | + | ++ | ++ | ND | ND | ND | (active site) |
| SMAB117 | rSMOX | 111 | - | - | - | - | - | + | ND | + | ND | - | ND | - | - | - | + | - | - | - | - | ND | ND | ND | |
| SMAB27 | rSMOX | 45 | - | - | - | + | + | + | ND | + | II | ++ | ND | - | + | + | + | - | + | +++ | ++ | + | ++ | + | C-terminal peptide |
| SMAB9 | LL1 | 20 | + | - | ND | ND | ND | + | ND | ND | IV | ++ | ++ | - | + | ++ | + | - | ++ | ++ | - | ++ | ++ | ++ | Long loop PEIEPR |
| SMAB16 | LL1 | 34 | + | - | ND | ND | ND | + | ND | - | ND | ++ | ++ | - | + | ++ | + | - | ++ | ++ | - | ++ | ++ | ++ | Long loop PEIEPR |
| SMAB19 | LL1 | 31 | + | - | ND | ND | ND | + | - | - | IV | ++ | ++ | - | + | ++ | - | - | + | +++ | + | ND | ND | ND | Long loop PEIEPR |
| SMAB21 | LL1 | 38 | + | - | ND | ND | ND | + | ND | - | IV | ++ | ND | - | + | ++ | ++ | - | +++ | +++ | - | + | + | + | Long loop PEIEPR |

ND: Not determined. Values in parenthesis indicate inferred or tentative assignments. See text for details.

spermine conversion in the absence and in the presence of excess SMAB (Fig 2). Families I and IV inhibited activity marginally, and family II inhibited moderately except for SMAB25 that inhibited strongly. Family III almost completely inhibited rhSMOX, making it likely that these SMABs binds close to the active site.

## Heavy chain SMAB sequence analysis

To be able to gauge diversity among the different SMABs, we sequenced the heavy chain (HC) portions. Regions containing the HC CDR1-3 sequences and a corresponding cladogram are shown in Fig 3. The HC CDR's of antibodies raised against the LL1 peptide (family IV; SMAB# 116, 119, 121, 39) are identical except for two minor amino acid substitutions in SMAB121. We note that these SMABs might still differ in reactivity due to possible differences in their Light Chain CDR sequences. Also, SMAB38 and SMAB103 in family III differ by only 1 residue in the HC. In family II, SMAB10 and SMAB34 are similar to each other (28/35 identical residues). SMAB16 and SMAB26 were not tested in epitope binning, but they share homology with SMAB10 and SMAB34 and are thus tentatively assigned to family II. A significant similarity is also observed between SMAB5 and SMAB101 of family I and may bind to the same epitope.

## Western blotting

To determine which SMABs recognize linear hSMOX epitopes and are suitable for Western Blot applications, we performed denaturing WB using rhSMOX, rhPAOX, and lysates of non-stimulated and BENSpm stimulated A549 cells. Relative staining scores obtained by visual inspection are collected in Table 1. Examples of hSMOX positive and hSMOX negative blots are shown in Figs 4, S4 and S5. With SMABs that provided an hSMOX signal, the signal intensity differences between non-stimulated and stimulated A549 cells was 8-13-fold. None of the SMABs recognized rhPAOX. As expected, none of the tested SMABS provided hSMOX bands

### A

| SMAB | Family | CDR1 | CDR2 | CDR3 |
|------|--------|------|------|------|
| 2 | I | GFSLSSYAIIW | EWIGIINGDGIRY | RGD--NWGTGHAPSI |
| 5 | I | GFSLSSYNMGW | EYIGFVSSIGSTY | RNRF---GYVYGEDI |
| 101 | I | GFSLSSYGVSW | EYIGFVSSIGSTY | RSRF---GYVFGEDI |
| 3 | II | GFSLSSNGVSW | EYIAFIGTTGITN | R-PGAFYGA-----I |
| 10 | II | GFSLNNYAISW | EWIGAINSYGTTY | R--HVDGL--YYSNI |
| 17 | II | GFSLSNYGMTW | EWIGFIGSGGSTY | R---GDGGGYVALNI |
| 25 | II | GIDLSSNAITW | EWIGIINSYGGTF | RVGTNSVGYAYA--L |
| 34 | II | GFSLGNYGVLW | EWIGAIGSSGTTY | RYV--DGLY--YSNI |
| 127 | II | GFSLSSYGVSW | EYIAFIGSAGYSN | R-PGAFYGA-----I |
| 38 | III | GFSLSSYAISW | EWIGVIDSGGNTY | SNYEAYGGAWPPYNI |
| 103 | III | GFSLSSYAISW | EWIGVIDSGSNTY | SNYEAYGGAWPPYNI |
| 112 | III | GFSLSSYGVIW | EWIGVIGPNGDAN | RG-------LGSLDI |
| 116 | IV | GFSLSTYAINW | EWIGAMGKSGSTY | R------GVFDKDGI |
| 119 | IV | GFSLSTYAINW | EWIGAMGKSGSTY | R------GVFDKDGI |
| 121 | IV | GFSLSSYAINW | EWIGAMGKSGSTY | R------GVFDKDGI |
| 39 | IV | GFSLSTYAINW | EWIGAMGKSGSTY | R------GVFDKDGI |
| 12 | SING | GFSLSSYGVSW | EWIGFINSYGSTY | REGYGYGGA-Y--NI |
| 16 | ND | GFSLSSNAISW | EWIGAINSYGSTY | R-GYSNGLY-Y-FNI |
| 26 | ND | GFSLSSNAISW | EWIGAINSYGSTY | R-GYINGLY-Y-FNI |
| 104 | ND | GFSLSSYGVSW | EWIGIIDYYGRTY | R-GRSDGDI-YALNI |
| 33 | ND | GFSLGSNAISW | EWIGAIGSSGTAY | REAGWNSFYPSYFNI |
| 28 | ND | GFSLSNYGVSW | EYIGIINNYGFTG | RAPYYH-PVVYGMDL |
| 117 | ND | GFSLYSYGVIW | ECIGTIGSSGSAY | RS--LFGTVGYAFNI |

### B

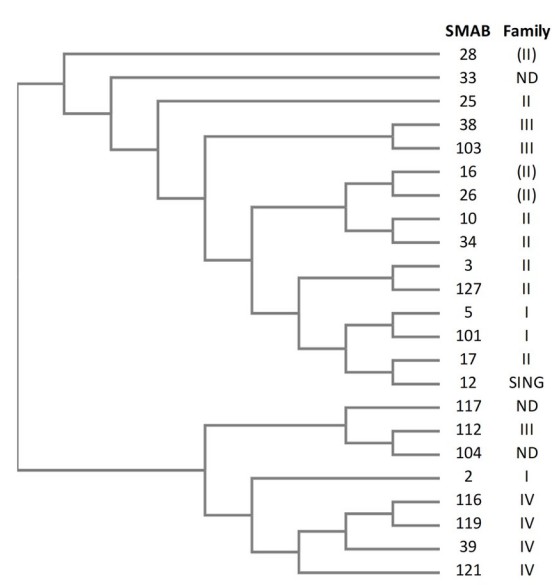

**Fig 3. Heavy chain SMAB sequence analysis.** (A) Heavy-chain CDR sequences and (B) a corresponding cladogram illustrating the similarity between SMABs. Family numbers in parenthesis correspond to tentative inferred assignments.

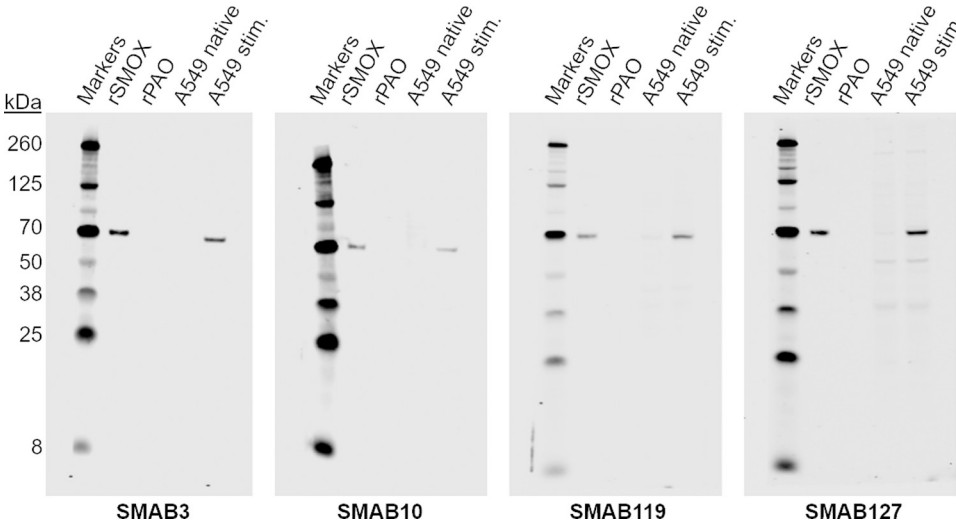

**Fig 4. SMABs in Western blot.** Application of selected SMABs in Western blot using control and BENSpm stimulated A549 cells. rhSMOX and rhPAOX were used as positive and negative controls, respectively. For the blots shown, rhSMOX bands were also visible for non-stimulated A549 cells upon increasing signal intensity (not shown). GAPDH was used as a loading control (see S4 Fig). Loading: 10 ug lysate, 1 ng rhSMOX, 1 ng rhPAOX.

in Raji cell lysate samples (not shown). From the SMABs generated using rhSMOX as the antigen, only those that recognized the CT peptide provided clear rhSMOX bands, indicating that the remainder of the rhSMOX SMABs recognize conformational epitopes. The four tested SMABs raised against the LL2 peptide all recognized purified rhSMOX and native hSMOX in (stimulated) A549 cell lysates. Of the SMABs providing intense hSMOX bands, clones SMAB3 (CT specific) and SMAB119 (LL1 specific) showed almost no background staining and thus appear most suitable for Western Blot applications. A selection of SMABs were also tested in WB against recombinant murine SMOX (rmSMOX; 95% identity with hSMOX) resulting in a positive signal for all hSMOX positive SMABs (Table 1 and S6 Fig).

## AlphaLISA immunoassay

For the detection and quantification of native hSMOX in complex biological sample matrices such as cell lysates, we developed a homogeneous immunoassay using the bead-based proximity assay platform AlphaLISA from PerkinElmer. This assay format utilizes donor and acceptor beads that, in our case, are each coated with a different antibody binding to the analyte. When the analyte is present, the donor and acceptor beads are brought into proximity resulting in a luminescence signal after excitation of the donor beads. This assay format is expected to have high specificity as signal generation depends on the simultaneous recognition of hSMOX by two different SMABs. We utilized anti-rabbit IgG donor beads for the binding of native SMAB, and anti-His tag acceptor beads for the binding of a HIS-tagged Fab fragment (Fig 5A). We selected 3 SMABs that recognize a conformational epitope on native hSMOX (SMAB# 2, 33, 26) to generate HIS-tagged Fab fragments by routine cloning and HEK293 expression. The resulting Fab fragments bound to immobilized rhSMOX with high affinity providing ELISA EC50 values 2 to 3-fold lower than the EC50 of the parent SMABs (not shown). The Fabs were tested in AlphaLISA in a checkerboard format against SMAB# 2, 10, 17, 26 and 33 in the presence and absence of non-tagged rhSMOX. The combination with the highest signal-to-noise ratio (Fab33 and SMAB2) was selected for follow-up. The concentrations of Fab33 and SMAB2 were optimized to 1 nM and 0.1 nM respectively. The final assay allows rhSMOX

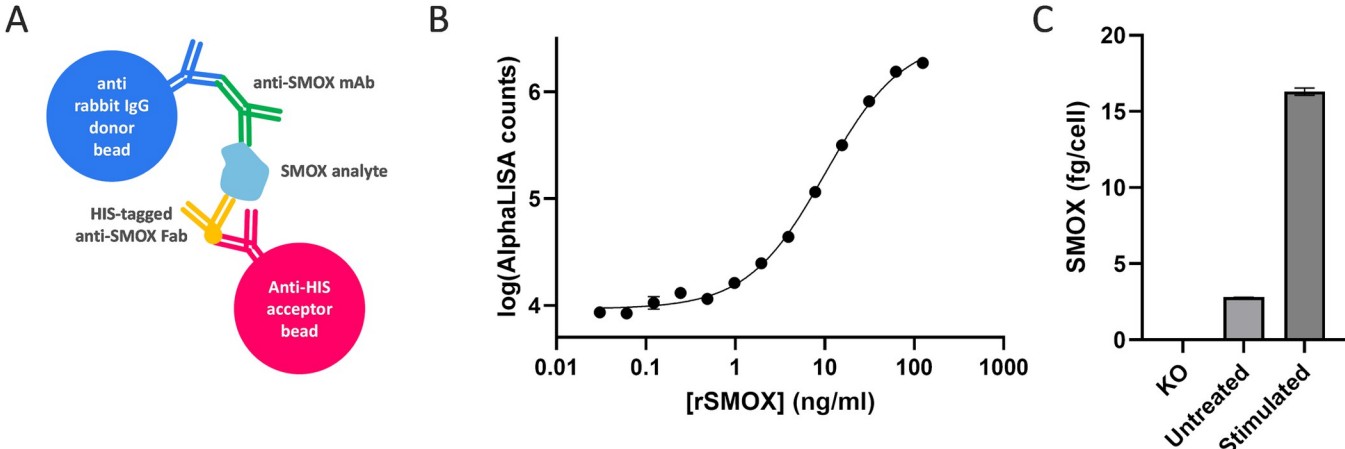

**Fig 5. SMABS in hSMOX detection AlphaLISA assay.** (A) schematic representation of the assay principle. (B) Example rhSMOX calibration curve with each [rhSMOX] in duplicate using the final assay format as explained in the main text (EC50 = 10.1 ng/ml; 95% confidence interval 9.0–11.3 ng/ml; $R^2$ = 0.997). Note that the symbol size is larger than the standard deviation of duplicate measurements for the majority of calibration points. (C) Use of the assay to quantify native hSMOX in A549 cells in which the hSMOX gene was knocked-out (KO), untreated A549 cells, and BENSpm stimulated A549 cells expressed in fg/cell. All data points represent the averages ± std of duplicate measurements.

quantification between 1 and 100 ng/ml (Fig 5B). The assay was utilized to quantify hSMOX in non-stimulated A549, BENSpm stimulated A549, and hSMOX knock-out A549 cell line lysates at an equivalent of 20,000 cells/well (Fig 5C). The knock-out cell line provided values below the LLOD demonstrating that the background signal is negligible in a complex biological sample matrix. As expected, [hSMOX] in BENSpm stimulated cells was higher than in non-stimulated A549 (16.3 vs 2.8 fg/cell; 5.8x increase).

## Immunofluorescence and immunohistochemistry on fixed cell lines

To select SMABs for in-depth performance testing using human tissues as described further below, we first performed immunofluorescence staining on all selected SMABs using non-stimulated A549 and BENSpm stimulated cells at ~75% and at full confluency. To avoid experimental variability, unstimulated and stimulated cells were cultured in the same 96-well plates, stained, and imaged together. Cells were formalin-fixed, permeabilized, stained with primary SMAB at four different concentrations, and counterstained using anti-Rabbit-IgG-AF488 and the nuclear stain DAPI. Images of all imaged wells are shown in S7 Fig, and selected images are shown in Fig 6. In non-stimulated A549 cells, significant staining was observed for a subset of cells indicating that hSMOX expression differs between individual cells. To quantify the staining, the blue (nuclei) and green (SMAB) channels of the wells with fully confluent A549 cells stained with 0.3 ug/ml SMAB were integrated and binned into four classes as reported in Table 1. Judging by the SMAB staining intensities, the induction by 10 uM BENSpm appears higher in fully confluent cells as compared to ~75% confluent cells. Whereas the hSMOX increase after BENSpm stimulation at 75% confluency is 5x-10x, the observed stimulated/non-stimulated staining ratio cells was more than 40-fold for SMAB# 5, 34, 101 and 103 at full confluency. The reasons for this difference were not investigated further. For the majority of SMABs, strong staining in stimulated cells was observed even at the lowest concentration tested (0.1 ug/ml), in line with the high affinities observed in ELISA (Table 1). The SMABs raised against the long-loop LL1 peptide showed strong nuclear staining in both non-stimulated and stimulated A549 cells (see Fig 6 for SMAB116). This might be related to the recognition of another SMOX isoform, e.g. isoform 5 (Genbank ID ABM01872.1) that has been

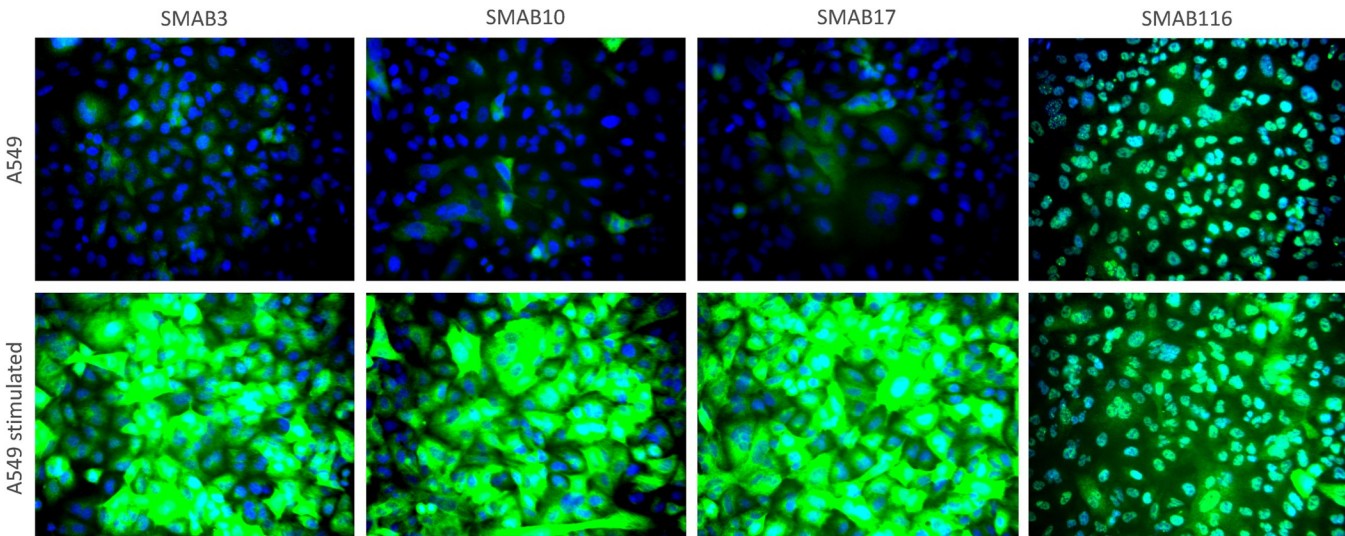

**Fig 6. SMABs in immunofluorescence.** Immunofluorescence images using 1 ug/ml SMAB to stain fully confluent non-stimulated (top) and BENSpm stimulated A549 cells (bottom). DAPI stained nuclei are shown in blue and the SMAB signal is shown in green.

described to be primarily localized to the nucleus and also contains the LL1 peptide sequence [13]. Yet, given that Isoforms 1 and 5 are highly similar (95% identity) and both contain the same C-terminal sequence [13], it would have been expected that some of the conformational or CT specific SMABs would also recognize isoform 5 and result in nuclear staining similar to the anti-LL1 SMABs. Alternatively, the nuclear LL1 SMAB staining might be due to the recognition of other proteins that contain epitopes similar to the hSMOX sequence PEIEPR (see above). Although this sequence as a whole appears to be unique for hSMOX, homologous sequences occur in e.g. cap methyltransferases (CMTR1 and PCIF1; sequence PEVEPR) and TP53 binding protein 1 (TP53BP1; sequence PEIEP) that are primarily localized to the nucleus.

Next, we performed immunohistochemistry on paraffin-embedded cells using a selection of SMABs (see Table 1 for SMABs tested) to serve as a model for tissue staining. In this series of experiments, next to unstimulated and stimulated A549 cells, we also imaged Raji cells that have very low basal SMOX expression [28] as confirmed by Western Blot (S1 Fig), to serve as a negative control. Relative intensity scores at 10 ug/ml SMAB are reported in Table 1. Representative images are shown in Fig 7. Consistent with the immunofluorescence results, the SMABs raised against LL1 peptide (e.g. SMAB116) as well as SMAB3 showed significant nuclear staining on Raji cells (see Fig 7 for SMAB116) and these SMABs were deselected for follow-up. SMAB10 and SMAB17 demonstrated a desirable profile (no Raji staining and clear contrast between stimulated and non-stimulated A549 cells) with SMAB10 showing somewhat better sensitivity.

In the next step, an IHC assay utilizing SMAB10 was further optimized on cell pellets (Fig 8A) and additionally tested to assess immunoreactivity in Raji and A549 tumor xenografts grown in nude mice. As expected, SMAB10 showed specific cytoplasmic immunoreactivity in A549 xenografts, but not in Raji control xenografts (Fig 8B).

To validate the use of SMAB10 and the developed staining protocol for staining human tissues, SMAB10 was first used to stain whole section samples of different human cancers followed by a tissue microarray surveying 47–49 samples of four common human malignancies and reference normal tissue (Fig 9A). The tissue microarray SMAB10 staining were binned

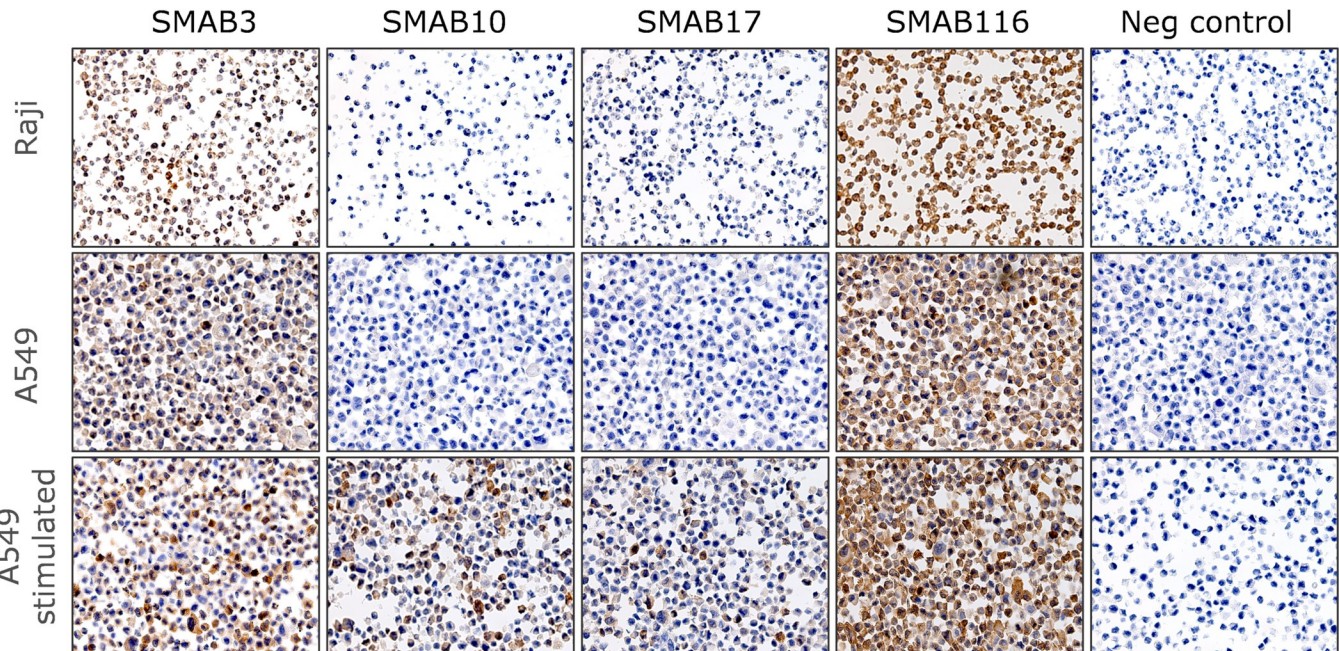

**Fig 7. SMABs in immunohistochemistry** Immunohistochemistry on paraffin-embedded Raji and A549 cells and A549 cells stimulated with BENSpm.

into 4 classes (negative or grade I-III) by visual inspection as shown in Fig 9B. In the sections tested, colon cancer tissue showed the greatest intensity and highest frequency of positive staining. Examples of positive tumor staining was also visible in breast and prostate cancer tissue, and least frequent and intense in lung cancer tissue (Fig 9B). In all cases, hSMOX staining appeared more frequent in cancer as compared to normal tissue, as previously reported [26,29,30]. Interestingly, we observed occasional instances of apical membranous staining as well as nuclear staining (S8 Fig).

## Discussion

Here we describe the generation and characterization of a series of new monoclonal antibodies recognizing human spermine oxidase, hSMOX. Spermine oxidase plays a crucial role in polyamine catabolism and homeostasis and has come into view as a potential drug target in the field of cancer prevention and interception, among others. The availability of selective and high-affinity monoclonal antibodies is essential to visualize, detect and/or quantify hSMOX in different experimental settings, thereby providing the tools to understand the complex biology of hSMOX, to validate its role in health and disease, and to aid in hSMOX targeted drug development. We found that the three commercial polyclonal antibody preparations we evaluated were unsuitable for our purposes. We chose to generate rabbit antibodies because of their general high specificity and affinity when compared to e.g. mouse antibodies[31]. A series of high-affinity mAbs were generated from rabbits immunized with peptide LL1 and native recombinant hSMOX. Antibodies SMAB3 and SMAB119 recognize linear epitopes, respectively, in the C-terminal sequence and in the sequence PEIEPR that is part of a long loop that occurs in hSMOX but not in the homologues enzyme hPAOX. These mAbs were found to be especially suitable in western-blot applications, especially with regard to the very low background signal in cell lysates. Antibodies in family III (SMAB# 38, 112, 103) were found to efficiently block hSMOX activity, presumably through binding to residues close to the active site, and these

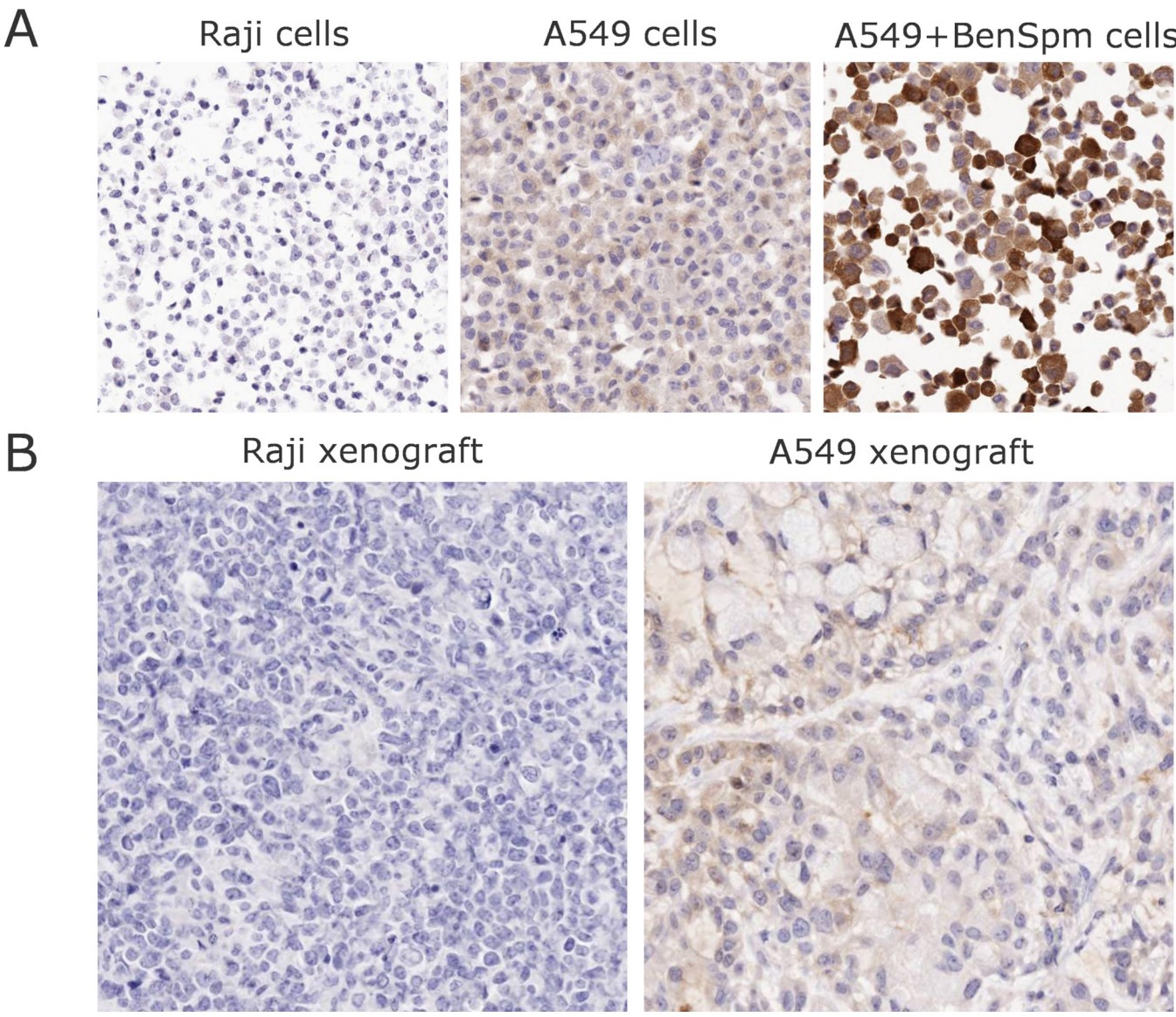

**Fig 8. Immunohistochemistry using SMAB10.** Images of hSMOX staining on paraffin embedded A549 cells (A) and mouse xenograft tissues (B).

mAbs would be suitable to selectively block native hSMOX activity in complex mixtures. The high-affinity binding to hSMOX by two mAbs simultaneously allows for the development of sensitive and highly selective assays to quantify native hSMOX in solution, as shown here by our AlphaLISA based assay employing SMAB2 and a Fab fragment from SMAB33. To understand the disease relevance of hSMOX, for example in cancer, and to visualize hSMOX in different human tissues or cell lines, antibodies suitable for immunofluorescence and immunohistochemistry are crucial. In immunofluorescence experiments using A549 cells, SMAB# 5, 34, 101 and 103 all combined low background signal with excellent contrast between non-stimulated A549 cells and A549 cells in which hSMOX expression was induced using BENSpm. For immunohistochemistry, we selected SMAB10 and show that this mAb is capable of detecting hSMOX in xenografts and human cancer tissues. SMAB10 staining appeared more frequent in cancer tissues as compared to normal tissues, in line with a possible

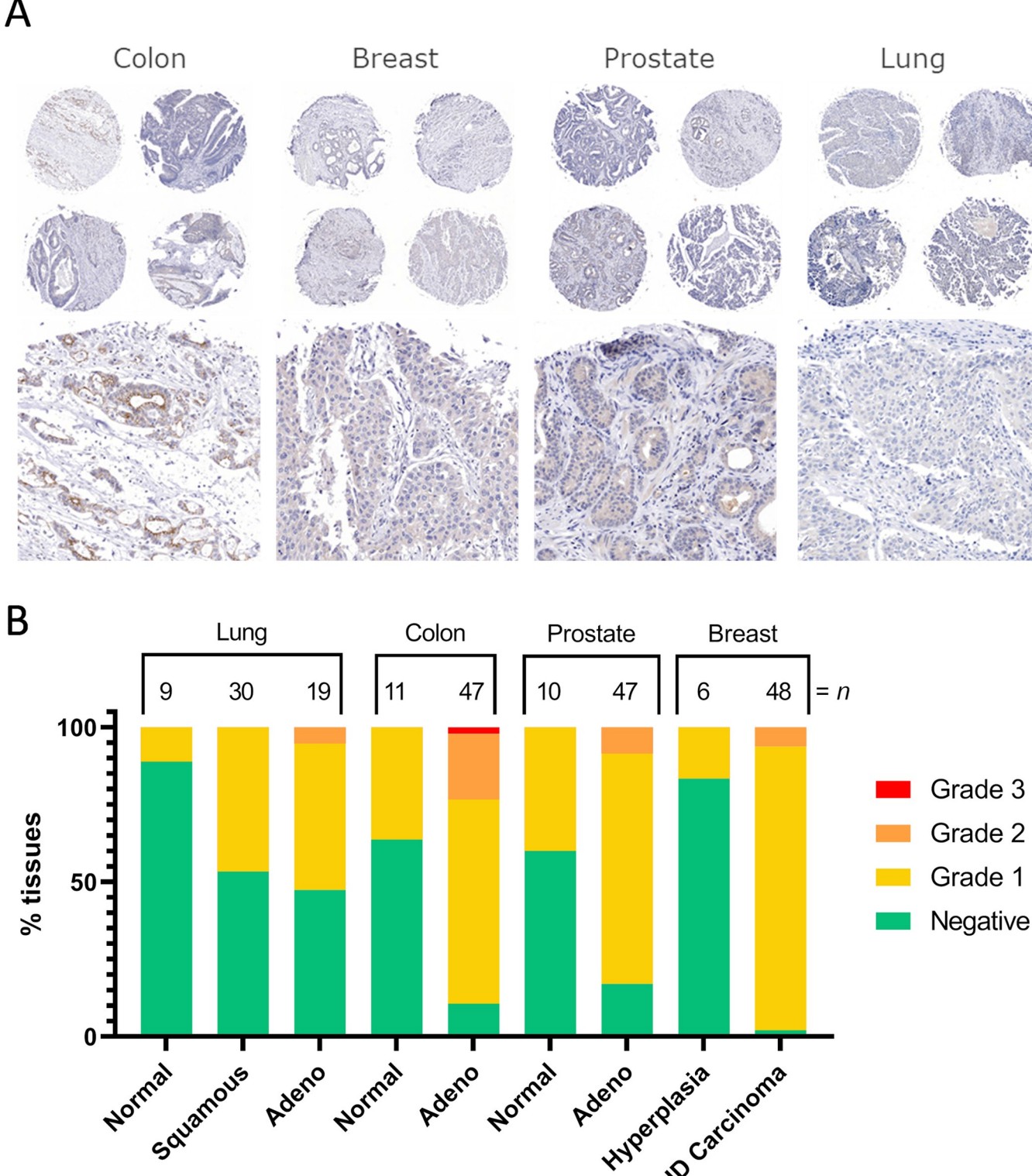

**Fig 9. Human tissue microarray.** (A) Representative tissue microarray immunohistochemistry images on 4 different human cancers. (B) hSMOX staining grading frequencies in tissue microarrays.

role of SMOX in cancer development [29,30]. In conclusion, the antibodies characterized here should meet the needs of most investigations involving hSMOX detection on gels, in solution, and in mammalian cells and tissues. Whereas polyclonal antibodies are characterized by lot-to-lot variability and are not suitable for some experimental settings, the mAbs described here can be reproducibly generated in large quantities from recombinant expression and are suitable for most common applications.

## Supporting information

**S1 Fig. Commercial polyclonal antibodies against hSMOX.** Western blot using commercial polyclonal antibodies probing their capacity to recognize rhSMOX and native hSMOX in different cell line extracts.
(TIF)

**S2 Fig. A549 immunofluorescence with commerical polyclonal antibodies.** Immunofluorescence of A549 cells and BENSpm stimulated A549 cells using three different commercial polyclonal antibody preparations at various dilutions. Nuclei (DAPI) are shown in blue, and hSMOX (α-Rabbit-IgG-AF488) is shown in green.
(TIF)

**S3 Fig. Sequence alignment of human spermine oxidase (hSMOX) and human polyamine oxidase (hPAOX).** The peptide sequences used for immunization are highlighted using the color coding of Fig 1 in the main text (NT: blue; SL: orange; LL2: purple; LL1 and LL2 overlap: yellow; LL1: cyan; CT: red).
(TIF)

**S4 Fig. Representative Western blots from Fig 4.** Western blots are overlaid with GAPDH loading control.
(TIF)

**S5 Fig. Representative blots from hSMOX monoclonal antibodies that were Western negative.** Faint bands visible near 70 kDa marker, nonspecific bands are similarly stained. GAPDH was used as a loading control (see S4 Fig). Loading: 10 ug lysate, 1 ng rhSMOX, 1 ng rhPAO.
(TIF)

**S6 Fig. Cross reaction of selected SMABs against rmSMOX.** Western blot assessing the reactivity of selected SMABs towards recombinantly expressed human and murine SMOX (1ug SMOX per lane; 1:1000 dilution of primary detection SMAB). For all samples, the order is: Lane 1 Marker, lane 2 Human SMOX, lane 3 Murine SMOX.
(TIF)

**S7 Fig. A549 cells immunofluorescence.** Immunofluorescence of fixed non-stimulated and BENSpm stimulated A549 cells at ~75% and full cell confluency using different SMABs at various concentrations. Nuclei (DAPI) are shown in blue, and SMAB (α-Rabbit-IgG-AF488) is shown in green.
(TIF)

**S8 Fig. Immunohistochemistry of colon cancer tissue.** Image illustrating apical membranous staining observed in colon cancer (left) and nuclear staining in normal prostate tissue (right).
(TIF)

**S1 Table. Statistical analysis for the AlphaLISA rhSMOX assay.** List of the row data and statistical analysis parameter.
(XLSX)

**S1 Raw images.**
(PDF)

## Acknowledgments

The authors thank Murray McKinnon, Paul Klatser, Els Brinkman and Michael Sharp for supporting this research and for helpful discussions and Kunhua Chen (Exonbio, San Diego, USA) where rabbit immunization was performed.

## Author Contributions

**Conceptualization:** Antonietta Impagliazzo.

**Data curation:** Armand W. J. W. Tepper, Gerald Chu, Jay H. Kalin.

**Formal analysis:** Armand W. J. W. Tepper, Gerald Chu, Patricia Molina-Ortiz.

**Investigation:** Armand W. J. W. Tepper, Gerald Chu, Vincent N. A. Klaren, Jay H. Kalin, Patricia Molina-Ortiz, Antonietta Impagliazzo.

**Methodology:** Armand W. J. W. Tepper, Vincent N. A. Klaren, Patricia Molina-Ortiz, Antonietta Impagliazzo.

**Project administration:** Antonietta Impagliazzo.

**Supervision:** Antonietta Impagliazzo.

**Validation:** Armand W. J. W. Tepper, Gerald Chu, Vincent N. A. Klaren, Jay H. Kalin.

**Visualization:** Vincent N. A. Klaren.

**Writing – original draft:** Armand W. J. W. Tepper, Jay H. Kalin, Patricia Molina-Ortiz, Antonietta Impagliazzo.

**Writing – review & editing:** Armand W. J. W. Tepper, Antonietta Impagliazzo.

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
