## [Decision Letter · Decision Letter 0]

7 Jan 2022

PONE-D-21-38029Development and characterization of rabbit monoclonal antibodies that recognize human spermine oxidase and application to immunohistochemistry of human cancer tissuesPLOS ONE

Dear Dr. Antonietta Impagliazzo,

Thank you for submitting your manuscript to PLOS ONE. After careful consideration, we feel that it has merit but does not fully meet PLOS ONE’s publication criteria as it currently stands. Therefore, we invite you to submit a revised version of the manuscript that addresses the points raised during the review process.

The editor declares a major revision decision, based on both reviewer comments, and taking in strong consideration the points that myself raised in the first submission.

In more details, reviewer #1 accomplished the benefit of your multiple monoclonal SMOX antibodies, but he invites the authors revising the list of references. The editor already invited the author to pose more attention on this point and ignoring it is certainly not a positive message for the editor and colleagues cited erroneously in the manuscript. Secondly, reviewer # 1 suggests to the authors providing evidences for a cross reaction with other model mammalian organisms beside human. It could not be an issue but it needs more experiments or comments.

Reviewer #2 recognizes the work as solid with appropriate amount of experiments. Unfortunately, still the description of the statistical analysis is completing missing, as well the editor asked upon the original submission. Providing a paragraph in MM and mentioning the appropriate statistical analysis used in each experiment means which kind of statistical tool have been applied (student-T test, ANOVA, median range, etc). Moreover, all the Material and Methods section should be revised and described in more details (as an example, in immune-histochemistry, how have negative control been determined?).

The editor still believe that your manuscript could be accepted, since both reviewers acknowledge the importance in the specific field of PA metabolism, but, once again, the editor invites primarily the corresponding author to work out on the points raised.

We look forward to receiving your revised manuscript.

Kind regards,

Roberto Amendola, Ph.D

Academic Editor

PLOS ONE

Journal Requirements:

Reviewers' comments:

Reviewer's Responses to Questions

**Comments to the Author**

1. Is the manuscript technically sound, and do the data support the conclusions?

Reviewer #1: Yes

Reviewer #2: Partly

2. Has the statistical analysis been performed appropriately and rigorously? 

Reviewer #1: N/A

Reviewer #2: No

3. Have the authors made all data underlying the findings in their manuscript fully available?

Reviewer #1: Yes

Reviewer #2: Yes

4. Is the manuscript presented in an intelligible fashion and written in standard English?

Reviewer #1: Yes

Reviewer #2: Yes

5. Review Comments to the Author

Reviewer #1: The current manuscript by Tepper et al. describes the development of multiple monoclonal antibodies that recognize human spermine oxidase SMOX). As correctly pointed out in the manuscript, SMOX appears to be an important enzyme associated with multiple pathological conditions including many cancers. Current data suggest that SMOX plays a role in the etiology of multiple epithelial cancers and as suggest may represent a rational target for chemoprevention or early intervention. As such, the availability of highly specific, SMOX antibodies that can be used for multiple types of analyses would benefit the continued study of the role of SMOX in disease.

The authors provide convincing data that they have developed sufficiently specific monoclonal antibodies to be used for Western blot, fluorescent microscopy, and IHC analyses of human SMOX. The data are presented in a clear and convincing manner and the tools described have been sufficiently tested for their selectivity in the various uses described.

There are no major concerns regarding the presented data, or the conclusions made by the authors. However, there are a couple of minor points that should be addressed.

1) The authors state that, “In conclusion, the antibodies characterized here should meet the needs of most investigations involving hSMOX detection on gels, in solution, and in mammalian cells and tissues.” However, the only data presented are results with human SMOX. It would be helpful to demonstrate that the antibodies also recognize mouse or other model mammalian organisms SMOX. Although, this shouldn’t be an issue, since mouse and human SMOX share extensive homology, experience with polyclonal antibodies suggests that not anti-human SMOX antibodies are useful for detecting mouse or other mammalian SMOX.

2) There appears to be several errors in the Reference section. There is duplication of references, incorrect authorship, missing journal citations, etc. These must be corrected.

Reviewer #2: Here it follows the evaluation of the MS (PONE-D-21-38029): “Development and characterization of rabbit monoclonal antibodies that recognize human spermine oxidase and application to immunohistochemistry of human cancer tissues” by Tepper et al.

This paper deals about the production of a panel of high-affinity rabbit monoclonal antibodies against various SMOX epitopes and selected antibodies for different molecular analysis as immunoblotting, protein quantification assays, immunofluorescence microscopy and immunohistochemistry. Authors have identified the crucial epitops to generate specific antibody against SMOX with different features. The work is solid and the manuscript well written. A great amount of experiments and data are reported. It is an important contribution towards understanding of polyamine metabolism supplying new tools to following in vitro and in vivo SMOX expression. In particular, SMAB10 antibody allows following SMOX expression in different cancer tissues.

The MS well adheres to the main points requested by Plos ONE journal policy but to major concerns need to be satisfied before acceptance.

Major points:

1) Statistical analysis is completing missing, please provide a paragraph in MM and mention the appropriate statistical analysis used in each experiment and specify it in figure legends.

2) A comparison with other commercial antibodies would be beneficial for Plos ONE readership as:

- Proteintech anti- SMOX Antibody 15052-1-AP |

- abcam anti-SMOX antibody ab213631

I recommend publishing of this MS on Plos ONE with the revisions outlined above.

6. PLOS authors have the option to publish the peer review history of their article (what does this mean?). If published, this will include your full peer review and any attached files.

Reviewer #1: No

Reviewer #2: No

---

## [Author Response · Author response to Decision Letter 0]

4 Mar 2022

Response to the reviewer of the manuscript:

Development and characterization of rabbit monoclonal antibodies that recognize human spermine oxidase and application to immunohistochemistry of human cancer tissues

Armand W.J.W. Tepper1; Gerald Chu2; Vincent N.A. Klaren1; Jay H. Kalin2; Patricia Molina-Ortiz3; Antonietta Impagliazzo1,4*

n.b. The reference are to the lines in the revised manuscript with track changes.

Reviewer #1 accomplished the benefit of your multiple monoclonal SMOX antibodies, but he invites the authors revising the list of references. The editor already invited the author to pose more attention on this point and ignoring it is certainly not a positive message for the editor and colleagues cited erroneously in the manuscript. Secondly, reviewer # 1 suggests to the authors providing evidences for a cross reaction with other model mammalian organisms beside human. It could not be an issue but it needs more experiments or comments.

• We apologize for the number of errors in the previous list of references. We have checked and revised the reference section and hopefully provided the correct author's names and format.

• Furthermore, we have reported the cross-reactivity of some of these antibodies with recombinant murine SMOX (Table 1 of the manuscript, line 389). We have added the figure of the WB in the supplementary information as well as an alignment of the human and murine sequence (line 382 and S6 Fig. line 384). We recognize that the quality of the blots is suboptimal, but unfortunately, we are unable to perform additional experiments due to the changes in circumstances. We hope, however, that this is convincing enough to illustrate the suitability of those mAbs for use in work with mouse models or tissues.

Reviewer #2 recognizes the work as solid with appropriate amount of experiments. Unfortunately, still the description of the statistical analysis is completing missing, as well the editor asked upon the original submission. Providing a paragraph in MM and mentioning the appropriate statistical analysis used in each experiment means which kind of statistical tool have been applied (student-T test, ANOVA, median range, etc). Moreover, all the Material and Methods section should be revised and described in more details (as an example, in immune-histochemistry, how have negative control been determined?).

• The statistical analysis is now clearly outlined in a paragraph added to the MM as the reviewer requested (row 253, Fig 5 legend at row 415). We have also added a S1 Table with the raw data and all the statistical parameters used in our analysis.

• Furthermore, additional detail has been added to the MM section regarding the AlphaLisa (row 156) and Immunohistochemistry (row 205-219; 223-233) with clarification on the negative control determination. In brief, we selected Raji cells as negative control because publicly available transcriptomic data indicate a low basal hSMOX expression in this cell line (row 278), consistent with our own data. Raji cells were used to produce paraffin-embedded cell and xenograft samples to serve as a negative control in IHC (Fig 7).

---

## [Decision Letter · Decision Letter 1]

1 Apr 2022

Development and characterization of rabbit monoclonal antibodies that recognize human spermine oxidase and application to immunohistochemistry of human cancer tissues

PONE-D-21-38029R1

Dear Dr.Antonietta Impagliazzo,

We’re pleased to inform you that your manuscript has been judged scientifically suitable for publication and will be formally accepted for publication once it meets all outstanding technical requirements.

Kind regards,

Roberto Amendola, Ph.D

Academic Editor

PLOS ONE

Additional Editor Comments (optional):

Reviewers' comments:

Reviewer's Responses to Questions

**Comments to the Author**

1. If the authors have adequately addressed your comments raised in a previous round of review and you feel that this manuscript is now acceptable for publication, you may indicate that here to bypass the “Comments to the Author” section, enter your conflict of interest statement in the “Confidential to Editor” section, and submit your "Accept" recommendation.

Reviewer #1: All comments have been addressed

Reviewer #2: All comments have been addressed

2. Is the manuscript technically sound, and do the data support the conclusions?

Reviewer #1: Yes

Reviewer #2: Yes

3. Has the statistical analysis been performed appropriately and rigorously? 

Reviewer #1: Yes

Reviewer #2: Yes

4. Have the authors made all data underlying the findings in their manuscript fully available?

Reviewer #1: Yes

Reviewer #2: Yes

5. Is the manuscript presented in an intelligible fashion and written in standard English?

Reviewer #1: Yes

Reviewer #2: Yes

6. Review Comments to the Author

Reviewer #1: The authors has sufficiently addressed my concerns and I believe if these monoclonal antibodies are made available to the polyamine community they will be of a major benefit.

Reviewer #2: Here it follows the evaluation of the MS (PONE-D-21-38029): “Development and characterization of rabbit monoclonal antibodies that recognize human spermine oxidase and application to immunohistochemistry of human cancer tissues” by Tepper et al.

In my opinion now, the manuscript is acceptable for publication.

7. PLOS authors have the option to publish the peer review history of their article (what does this mean?). If published, this will include your full peer review and any attached files.

Reviewer #1: No

Reviewer #2: No

---

## [Editor Report · Acceptance letter]

14 Apr 2022

PONE-D-21-38029R1 

Development and characterization of rabbit monoclonal antibodies that recognize human spermine oxidase and application to immunohistochemistry of human cancer tissues 

Dear Dr. Impagliazzo:

I'm pleased to inform you that your manuscript has been deemed suitable for publication in PLOS ONE. Congratulations! Your manuscript is now with our production department. 

Kind regards, 

on behalf of

Dr. Roberto Amendola 

Academic Editor

PLOS ONE